# Enteric Pathogen Testing Importance for Children with Acute Gastroenteritis: a Modified Delphi Study

Gillian A. M. Tarr,[a] Drew J. Persson,[a] Phillip I. Tarr,[b] Stephen B. Freedman[c,d,e]

[a]Division of Environmental Health Sciences, School of Public Health, University of Minnesota, Minneapolis, Minnesota, USA
[b]Division of Gastroenterology, Hepatology, and Nutrition, Department of Pediatrics, Washington University in St. Louis School of Medicine, St. Louis, Missouri, USA
[c]Departments of Pediatrics and Emergency Medicine, Cumming School of Medicine, University of Calgary, Calgary, Alberta, Canada
[d]Sections of Pediatric Emergency Medicine and Gastroenterology, Alberta Children's Hospital, Calgary, Alberta, Canada
[e]Alberta Children's Hospital Research Institute, Alberta Children's Hospital, Calgary, Alberta, Canada

**ABSTRACT** The application of clinical diagnostics for gastroenteritis in children has implications for a broad collection of stakeholders, impacting clinical care, communicable disease control, and laboratory utilization. To support diagnostic stewardship as gastroenteritis testing options continue to advance, it is critical to understand which enteropathogens constitute priorities for testing across stakeholder groups. Using a modified Delphi technique, we elicited opinions of subject matter experts to determine clinical and public health testing priorities. There was a high level of overall agreement (≥80%) among stakeholders (final round $n = 15$) that testing was important for *Campylobacter*, *Escherichia coli* O157 and other Shiga toxin-producing *E. coli*, *Salmonella*, *Shigella*, *Vibrio*, *Yersinia*, norovirus, and rotavirus. Immunocompromised children were identified as a special population that warranted the additional testing of three to four bacterial and parasitic targets. To support these clinical and public health testing priorities, diagnostic stewardship strategies can be employed, such as educating clinicians, developing new decision support tools, and using multiplex testing in concert with selective result reporting and annotation.

**IMPORTANCE** Children with diarrhea and vomiting who seek care can be infected with a wide variety of infectious agents. This study reports findings from a survey of clinical, public health, and laboratory subject matter experts on the infectious agents that are most important to test for. The majority agreed on the importance of testing children likely infected with several bacterial agents, as well as two common viruses. Although confirming a child is positive for a viral agent is unlikely to change clinical care, participants noted the importance of monitoring these viruses for public health purposes. To avoid over-testing children, however, these results should be used to support diagnostic stewardship strategies and design new decision support tools.

**KEYWORDS** acute gastroenteritis, diagnostic stewardship, decision support, enteric pathogen

Address correspondence to Gillian A. M. Tarr, gtarr@umn.edu, or Stephen B. Freedman, stephen.freedman@ahs.ca.

The authors declare a conflict of interest. G.A.M.T.: none. D.P.: none. P.I.T.: serves as a member on the Data Safety Monitoring Board of Inmunova. S.B.F.: none.

Technical advances in diagnostics for gastrointestinal pathogens have outpaced the development of best practices that balance the needs of the many stakeholders invested in the conduct and output of clinical gastroenteritis testing (1). Multiplex molecular panels have made it possible to receive rapid results on more than a dozen pathogens simultaneously. However, the tenets of diagnostic stewardship caution against overuse of such testing (2, 3). Healthcare organizations have several tools at their disposal to promote good gastroenteritis testing practices, such as provider education (4), criteria for approved use coupled with computerized order entry decision support (5), and selective result reporting (6). The foundation of the good testing practices being advanced by these tools is an understanding of which enteropathogens

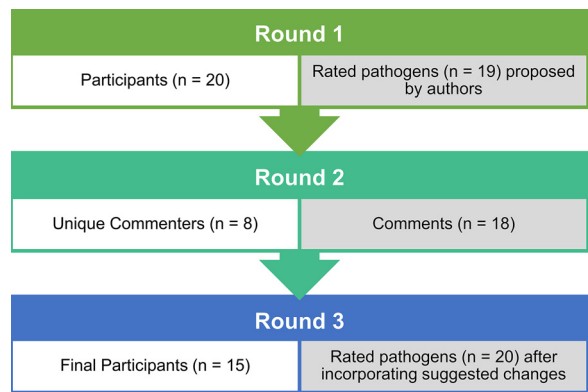

**FIG 1** Flow diagram of the Delphi process indicating the number of participants per round, and either the number of pathogens they were asked about (rounds 1 and 3) or the number of comments (round 2).

constitute priorities for testing, an uncertain target that continues to evolve as our knowledge of the causes of gastroenteritis expands.

In children with vomiting and/or diarrhea of presumed infectious origin, the timely identification of certain enteric pathogens may be critically important to inform treatment and public health responses (7). However, guidelines for testing children with gastroenteritis are inconsistent (8), and testing practices do not reflect any particular guideline, and, in fact, might be counterproductive (9). Bacterial and parasitic enteropathogens are frequently prioritized for testing, but some are of more concern than others given variations in severity, communicability, ability to treat, and meaningful association with disease. Antimicrobial treatment is only recommended for a subset of these pathogens and under select conditions (10, 11), as it has limited usefulness in uncomplicated cases and may lead to worse outcomes for children infected with Shiga toxin-producing *Escherichia coli* (STEC) (12). Even though there are no specific treatments, the detection of viral pathogens can also have value. Norovirus surveillance has been recognized as important for outbreak control and in anticipation of a potential norovirus vaccine (13). Data from the National Outbreak Reporting System (NORS) has also demonstrated the burden of non-norovirus viral outbreaks, arguing for broader testing and reporting (14). Moreover, the finding of norovirus can often explain vomiting without diarrhea, thereby obviating additional diagnostic interventions such as contrast studies.

The set of pathogens that should be prioritized for diagnostic testing has implications for clinicians who need to decide whether to order a test, clinic and laboratory leadership who determine which types of testing to support, clinical microbiology directors who oversee resource utilization, disease control specialists who rely on reporting of laboratory-confirmed cases for notifiable disease surveillance, and public health laboratory personnel who require particular types of submissions from clinical laboratories for additional notifiable disease testing (15–17). An understanding of these pathogens can also guide the development of new decision support tools designed to improve diagnostic stewardship (2, 3). Here, we provide a list of enteropathogens that gastroenteritis content experts representing the stakeholder groups listed above agreed should be prioritized for testing in children with gastroenteritis seeking medical care.

## RESULTS

Of the 56 subject matter experts (SMEs) invited, 20 (36%) responded in round 1, eight (14%) unique individuals made 18 comments in round 2, and 15 responded (27%) in round 3 (Fig. 1). Taking into account all stakeholder roles selected by participants, round 3 included 10 experts in gastroenteritis, nine clinicians, five medical laboratory professionals, one public health practitioner, and one field investigator (Table 1).

Regarding the 19 original pathogens in round 1 (Fig. 1), participants suggested the

**TABLE 1** Participant self-identified stakeholder roles in rounds 1 and 3

| Stakeholder role | Round 1 Total (as primary role) | Round 3 Total (as primary role) |
|---|---|---|
| Clinician | 12 (8) | 9 (7) |
| Medical laboratory professional | 5 (4) | 5 (3) |
| Public health practitioner | 5 (5) | 1 (1) |
| Acute gastroenteritis expert | 15 (3) | 10 (3) |
| Other (specified: field investigator) | 0 | 1 (1) |
| Total participants | 20 | 15 |

addition of five bacteria, seven parasites, and one virus; one bacterial pathogen, *Plesiomonas*, and one parasitic pathogen, *Cyclospora*, were suggested by >1 participant (Appendix 3 in the online supplemental material) and subsequently added to the round 3 survey. *Aeromonas* and non-40/41 adenovirus both had <50% overall agreement (i.e., support for priority testing) for testing in round 1 (Table 2). After round 2 discussion, *Aeromonas* was removed from the survey in round 3, and non-40/41 adenovirus was left on the survey because of its potential to alter clinical management in immunocompromised (IC) patients (Appendix 4 in the online supplemental material). Based on round 1 and 2 feedback, the clinical management domain was also differentiated based on immune status in round 3.

**High overall agreement for testing for most bacteria.** 10 bacterial enteropathogens were included in round 3, and ≥80% of participants agreed that seven of these were important to prioritize for testing for clinical management and/or public health purposes (Table 2). Agreement was high in both domains for STEC, including *E. coli* O157, *Salmonella*, and *Shigella*. Agreement was high for the importance of *Campylobacter* testing for clinical management but not public health purposes; however, when round 1 and 3 responses from only those participants identifying their stakeholder role as public health practitioner were examined, 100% indicated testing for *Campylobacter* was important for public health purposes (Table S1). One SME emphasized the public health importance of *Campylobacter* as, "Clustering of cases and point source might be identified for outbreak investigation, control, and prevention," (Appendix 5 in the online supplemental material). *Vibrio* and *Yersinia*, the former primarily for public health purposes and the latter for a mix of public health and clinical purposes, completed the list of seven bacterial enteropathogens that SMEs prioritized for testing.

**All participants prioritized testing for norovirus and rotavirus.** Overall agreement for the importance of norovirus and rotavirus testing was 100% in round 3 (Table 2). The high agreement for norovirus and rotavirus testing was driven by >90% agreement about the importance of testing for public health purposes. These two agreement measures are among only five that increased ≥25% between round 1 and round 3, the overall level of agreement for rotavirus being one of the others. Support for norovirus and rotavirus testing for public health purposes among public health practitioner stakeholders was largely concordant with the results from all participants (Table S1). No other virus reached a high level of overall or domain-specific agreement, making norovirus and rotavirus the only two viruses SMEs prioritized for testing.

**Parasitic enteropathogens were prioritized for testing with moderate overall agreement.** Participants did not agree at a high level on the importance of testing for any of the four parasitic enteropathogens we included in the round 3 survey (Table 2). Overall agreement for both *Cryptosporidium* and *Giardia* testing dropped from 90% in round 1 to 79% in round 3. *Cryptosporidium* prioritization was driven by its importance for public health purposes. For example, one participant commented, "Outbreaks do occur and certain high risk areas exist . . ." (Appendix 5 in the online supplemental material). Consistent with this, 100% of public health practitioners across rounds 1 and 3 indicated *Cryptosporidium* testing was important for public health purposes (Table S1). Similarly, there was greater agreement among clinicians regarding the importance of *Giardia* testing for clinical management, at 89% (Table S2), than among all participants. The stakeholder role-specific results suggest that *Cryptosporidium* and *Giardia* should be considered as testing priorities.

**TABLE 2** Percent agreement among all participants for recommendation of testing of enteropathogens in rounds 1 and 3 for clinical management, by immune status, and public health purposes

| Pathogen | Clinical management | | | Public health | | Overall[a] | |
|---|---|---|---|---|---|---|---|
| | Round 1 | Round 3 – IC[e] | Round 3 – NIC[f] | Round 1 | Round 3 | Round 1 | Round 3 |
| *Aeromonas* spp. | 15% | -[g] | - | 0% | - | 15% | - |
| *Campylobacter* spp. | 100%[b] | 93% | 87% | 80% | 73% | 100% | 87% |
| *C. difficile* | 95% | 87% | 67% | 45% | 29% | 100% | 67% |
| *E. coli* O157 | 95% | 93% | 87% | 90% | 100% | 95% | 100% |
| STEC[c] | 95% | 100% | 93% | 85% | 93% | 95% | 93% |
| ETEC[d] | 45% | 47% | 40% | 35% | 53% | 55% | 53% |
| *Plesiomonas* spp. | - | 29% | 29% | - | 29% | - | 36% |
| *Salmonella* spp. | 100% | 93% | 87% | 85% | 100% | 100% | 100% |
| *Shigella* spp. | 95% | 100% | 93% | 85% | 100% | 95% | 100% |
| *Vibrio* spp. | 65% | 73% | 67% | 70% | 87% | 85% | 87% |
| *Yersinia* spp. | 80% | 87% | 73% | 65% | 73% | 80% | 80% |
| *Cryptosporidium* spp. | 80% | 86% | 64% | 85% | 79% | 90% | 79% |
| *Cyclospora* spp. | - | 71% | 36% | - | 71% | - | 71% |
| *E. histolytica* | 70% | 79% | 71% | 50% | 57% | 75% | 71% |
| *Giardia* spp. | 85% | 86% | 79% | 65% | 57% | 90% | 79% |
| Adenovirus 40/41 | 70% | 71% | 43% | 35% | 43% | 75% | 64% |
| Adenovirus non-40/41 | 15% | 50% | 0% | 10% | 21% | 20% | 21% |
| Astrovirus | 40% | 29% | 14% | 25% | 36% | 50% | 43% |
| Norovirus | 65% | 71% | 50% | 70% | 100% | 80% | 100% |
| Rotavirus | 60% | 79% | 57% | 65% | 93% | 75% | 100% |
| Sapovirus | 50% | 43% | 29% | 35% | 64% | 60% | 71% |
| All negative | 65% | 69% | - | 20% | 8% | 70% | 69% |

[a]Overall agreement was calculated as the proportion of participants indicating a pathogen was important for either/both non-IC clinical management and/or public health purposes (i.e., the proportion of participants indicating importance for at least one of these purposes).
[b]Dark grey shading indicates a high level of agreement, ≥80%; light grey shading indicates a moderate level of agreement, ≥60% to <80%.
[c]STEC, Shiga toxin-producing *E. coli*.
[d]ETEC, enterotoxigenic *E. coli*.
[e]IC, immunocompromised.
[f]NIC, nonimmunocompromised.
[g]-, NA.

**Clinical management of IC children increases the pathogens to prioritize.** The importance of considering a child's immune status emerged as an important theme in round 1 comments and round 2 discussions, particularly in relation to adenovirus and astrovirus. In round 3, agreement regarding testing for the clinical management of IC children was a median of 10% higher (25th, 75th quartiles: 7%, 21%, respectively) than agreement for testing for general clinical management. Of the 11 pathogens that did not have high overall agreement in round 3, participants had a high level of agreement about the importance of testing for *C. difficile*, *Cryptosporidium*, and *Giardia* for the clinical management of IC children (Table 2), although these differences were not statistically significant (Table S3). As such, in this clinical context, these three should be considered as possible additions to the already-identified seven bacteria and two viruses as testing priorities. In round 3, clinicians also agreed at 89% on the importance of testing for *E. histolytica* for IC children, but they had slightly lower agreement than all participants about the importance of testing for *C. difficile* (Table S2).

**Disagreement existed regarding utility of testing to inform differential diagnosis.** For the several pathogens that received only low or moderate levels of agreement, a common theme among comments was that their testing could be useful in informing the differential diagnosis of a child with gastroenteritis. In round 1, one of the three participants who prioritized testing of *Aeromonas* commented, "Excluding it often enables us to move forward with other diagnoses, such as inflammatory bowel disease," (Appendix 3 in the online supplemental material). The concept emerged in round 2 in reference to ETEC, astrovirus, and sapovirus, with an acknowledgment that testing wouldn't change clinical management but could inform the differential diagnosis (Appendix 4 in the online supplemental material). Multiple participants reiterated this theme in relation to all or almost all

the viral enteropathogens in round 3 (Appendix 5 in the online supplemental material), none of which had high agreement for clinical management (Table 2).

Clarifying the diagnosis was also the intent behind the question regarding testing when it is suspected a child may be negative for all pathogens, which had moderate agreement among all participants at 69% (Table 2) but 88% agreement specifically among clinicians (Table S2). One clinician explained the importance thus: "Yes definitely for children being admitted to hospital with diarrhea and other conditions that may be causal (e.g., GI patients) or those with compromised immune systems (e.g., HIV, oncology)," (Appendix 5 in the online supplemental material). Illustrating the trade-offs, another clinician, who still indicated agreement with this testing, commented, "I wish. However, pan-pathogen testing elicits results that sometimes engender more anxiety than clarity, and encourage unnecessary treatment."

## DISCUSSION

There is a pressing need to understand testing priorities for children with gastroenteritis, both to guide policies supporting diagnostic stewardship within health care organizations and to develop new decision support tools. There was a high level of overall agreement among the diverse stakeholders we surveyed that testing was important for *Campylobacter*, *E. coli* O157 and other STEC, *Salmonella*, *Shigella*, *Vibrio*, *Yersinia*, norovirus, and rotavirus. Managing the care of an IC child could add three to four enteropathogens to those prioritized for testing based on overall agreement, which is consistent with gastroenteritis guidelines and provider surveys that rank IC status as a top reason for stool testing (10, 11, 18).

There was high agreement across both public health and clinical domains for *E. coli* O157, STEC, *Salmonella*, and *Shigella*. Illnesses associated with these pathogens, as well as *Campylobacter*, are of concern because of their potential for severe outcomes and associations with outbreaks. These bacteria are also often marked by frequent bloody or mucoid stools, fever, and/or abdominal pain (19, 20), and the consistencies in their clinical profiles can be used to guide testing. Indeed, bloody diarrhea, fever, severe presentation, and a concurrent outbreak are all included in existing recommendations as reasons to consider stool testing, so the prioritization of these pathogens is not surprising (10, 11).

Norovirus, the most common cause of medically attended acute gastroenteritis (21, 22), and rotavirus stood out as the only viruses prioritized for testing, driven by >90% agreement on testing for public health purposes. One respondent summarized that norovirus public health testing was, "Important for outbreaks, vaccine development, and monitoring," reflecting themes mentioned by several others and echoing those discussed in the literature (13). Similarly, monitoring current strains and the effectiveness of the rotavirus vaccine, which has been available in the United States since 2006, were given as reasons for prioritizing rotavirus testing. Strain monitoring may include tracking vaccine-derived rotavirus, which has not been causally linked to acute gastroenteritis (23), but is detected by standard rotavirus assays (24, 25). National surveillance systems such as the National Respiratory and Enteric Virus Surveillance System (NREVSS), which has provided substantial insight into rotavirus dynamics following vaccine introduction (26, 27), rely on the testing practices at the participating hospitals and clinics. However, current diagnostic practices do not reflect the public health importance of testing when norovirus and rotavirus are likely causes of a child's etiology. A survey of physicians from major American medical associations revealed that only 5% to 20% of pediatricians or family practice physicians reported ordering norovirus antigen, recommended only for use in an outbreak context (28), or PCR tests, and 25% to 40% had ordered a rotavirus antigen test (18). In that survey, the presence of vomiting, a common norovirus symptom, was most likely to reduce the likelihood of a physician ordering a stool test of any kind. If norovirus or rotavirus testing is to be prioritized, education will be needed to overcome these types of practice patterns.

Four of the prioritized pathogens would not have been identified had we not considered the importance of testing for public health purposes. Bridging the gap between

public health and clinical indications is important given the reliance of disease surveillance and outbreak detection on results generated from clinical testing (29), making public health considerations important for front-line clinicians. The prioritization of relatively rare pathogens like *Vibrio* and *Yersinia* is not likely to pose a large testing burden. However, as the most common cause of gastroenteritis, testing every likely case of norovirus could potentially overwhelm clinical laboratories without adding a proportional public health benefit. Awareness of community and patient characteristics that modulate the public health
importance of a potential case can inform the testing decision. These may include the existence of ongoing outbreaks, current disease levels in the community, point in the norovirus season, opportunities for spread (e.g., childcare attendance), or other local factors.

It is generally important in Delphi processes to include a diversity of opinions (30), and we considered it especially so in this case because of the many stakeholder groups and potentially broad implications of testing decisions. This approach aligns with the commonly accepted approach to include representatives from multiple clinical and nonclinical areas on laboratory test utilization committees and diagnostic stewardship teams (2, 31, 32). The SMEs we invited included individuals from clinical, laboratory, and public health disciplines, who in many cases had documented expertise in pediatric gastroenteritis. However, the low number of SMEs identifying as public health practitioners, particularly in the final round, may have limited the desired diversity of the participant pool. It is possible some individuals we invited to represent public health considered themselves better described by stakeholder roles other than practitioner, such as content expert or field investigator, because of the nature of their day-to-day work. Additionally, several of the individuals invited from clinical or laboratory disciplines would have had knowledge of public health; they conduct research that informs public health policy and practice, oversee public health laboratories, and collaborate closely with public health departments on a wide range of initiatives. While we cannot know which individuals responded and their exact list of qualifications due to anonymity, >70% of participants in rounds 1 and 3 indicated two stakeholder roles (Table 1), suggesting broad subject matter expertise. We also sought to assess the impact of the limited number of public health practitioner respondents. Based on our subanalysis of only public health practitioners, greater representation of this group may have yielded higher overall agreement for *Cryptosporidium* and potentially *Cyclospora*. However, high overall agreement was still achieved for most other pathogens for which public health domain agreement diverged between the full participant group and public health practitioners, leaving the majority of our results unchanged.

Although our primary goal was to determine agreement from a diverse pool of SMEs, the perspectives of those practicing within a particular domain may be considered to have greater weight in some situations. In addition to the nine enteropathogens that had high overall agreement from all participants and high agreement for *Cryptosporidium* among public health practitioners, clinicians also agreed that *Giardia* is a pathogen that should be tested for purposes of clinical management. We believe *Cryptosporidium* and *Giardia* should be evaluated as potential testing priorities based on local and organization-specific circumstances.

Multiplex molecular panels could potentially facilitate testing for all of the pathogens identified as important across the clinical and public health domains, in addition to pathogens some SMEs identified as of interest to clarify the differential diagnosis. Such pan-pathogen testing raises several concerns, as the unrestricted use of multiplex panels has been associated with low-value care due to increased costs and laboratory utilization with few benefits (33). Testing for a high number of pathogens can additionally complicate reimbursement for patients and providers if payment is based on the number of targets or positive findings (34). However, targeted use in children with symptoms and history that potentially match several pathogens could improve their utility and reduce the likelihood of incidental findings (35). If used, multiplex panels should be coupled with established diagnostic stewardship strategies (36), including

annotation of pathogens not likely to benefit from antimicrobial treatment (e.g., STEC [12, 37] and *Salmonella* [11]). In addition, selective suppression of results likely to indicate colonization (e.g., *C. difficile* in children <2 years old [38, 39] and EPEC in high-income settings [40, 41]) can prevent unnecessary treatment of these conditions. Suppression is ideally done at the instrument level to avoid putting laboratory personnel in the position of having information that is not made available to the clinical team. Thus, the decision regarding results suppression is complex and should be approached with particular caution.

This study was limited by the low response rate. Our 20 (round 1) and 15 (round 3) participating SMEs are not out of the ordinary among Delphi-like studies; some of the larger methodologic reviews of the Delphi literature reported a median of 17 invited experts (42) and 11 to 25 participants in the final round (43). A larger, more representative sample would have provided a greater diversity of opinions and may have yielded greater stability in our results between rounds for those pathogens that were not the focus of intense discussion. Particularly, agreement for the public health importance of diseases like *Campylobacter* and *Yersinia* may have been higher. It is also possible that the norovirus and rotavirus round 3 results would have been closer to their round 1 results, though this is difficult to judge because participants could have been influenced by survey feedback. This study was also limited by current diagnostic technology, testing norms, and knowledge of disease burden. Pathogens such as *Aeromonas*, *Plesiomonas*, *Cyclospora*, and astrovirus are rarely tested for and not included on the majority of multiplex gastrointestinal pathogen panels. The same is true for most pathogens that participants had suggested be added to the survey in round 1, such as *Edwardsiella* and human bocaviruses. It is possible that if these pathogens were more frequently tested for, they would be more frequently observed, and our participants would have considered them of higher importance.

SMEs identified seven bacterial and two viral enteropathogens as testing priorities for children with gastroenteritis, with potential additional bacterial and parasitic targets for IC children. However, multiplex diagnostics are not the only way to approach this list of testing priorities. The testing priorities we identified should be integrated as part of larger gastroenteritis diagnostic stewardship efforts focusing on provider education and decision support tools that could increase the pretest probability of any testing conducted (36). Machine learning techniques can be leveraged to develop tools that use symptom profiles of pathogens or pathogen groups to guide clinicians toward the most likely etiology (44). To effectively accommodate needs when working with specific populations (e.g., IC children), incorporate public health considerations, and support recommendations from increasingly detailed gastroenteritis guidelines (11), decision support tools also need to be programmable with different community and patient characteristics. Only with such flexible solutions will we see alignment between clinical and public health testing priorities, diagnostic stewardship, and actual gastroenteritis testing practices.

## MATERIALS AND METHODS

We ascertained the level of agreement about which pathogens should be prioritized for testing using a Delphi technique with three predetermined rounds (30). Adhering to classical Delphi practices, rounds 1 and 3 involved administration of a standardized questionnaire that was adapted between rounds, and participants were anonymous and received the results of each previous round. Departing from the standard Delphi technique, round 2 was a facilitated online discussion. Additionally, our objective was not to obtain consensus, but simply to ascertain the level of agreement among experts, a type of Delphi technique more commonly used outside the health sciences (30).

We identified SMEs for inclusion from the United States and Canada using a multitiered process. First, we sought to represent the diverse perspectives from major stakeholders (30) affected by clinical testing decisions. Next, we identified specialties and subspecialties of clinicians that would be impacted by or have expertise in children with gastroenteritis. We identified professional organizations aligned with the targeted specialties and, where possible, their relevant committees, and invited clinicians associated with them. We selected public health representatives with known content expertise in gastroenteritis and/or foodborne pathogens from the United States Centers for Disease Control and Prevention, Public Health Agency of Canada, state and provincial health departments, and academic public health centers. We identified a mix of clinical microbiology and public health laboratory

representatives. For all three categories, individuals known based on the literature to possess content expertise in gastroenteritis and its diagnostic stewardship were included whenever possible. In total, we invited the following groups of SMEs to participate in the study: clinical experts ($n = 29$) including physicians and nurse practitioners from family practice, general pediatrics, pediatric emergency medicine, pediatric infectious diseases, pediatric gastroenterology, and infection prevention and control; public health professionals ($n = 14$); and laboratory managers/directors ($n = 13$). Invitations were sent via email. Participation was anonymous; all participants were invited to participate in each round regardless of participation in the previous round. The University of Calgary Conjoint Health Research Ethics Board approved this study.

Rounds 1 and 3 consisted of Qualtrics surveys (Appendix 1 and 2 in the online supplemental material). Participants indicated their primary and secondary stakeholder roles, whether they felt it was important to test for each pathogen listed for clinical and/or public health reasons, and any qualitative comments regarding the importance of testing for each pathogen. Results from the previous round were provided to participants after each round. Round 1 included 19 pathogens: *Aeromonas*, *Campylobacter*, *Clostridioides difficile*, Shiga toxin-producing *Escherichia coli* (STEC), *E. coli* O157, enterotoxigenic *E. coli* (ETEC), *Salmonella*, *Shigella*, *Vibrio*, *Yersinia*, *Cryptosporidium*, *Entamoeba histolytica*, *Giardia*, adenovirus types 40/41, non-40/41 adenovirus, astrovirus, norovirus, rotavirus, and sapovirus. A question regarding the importance of testing if a child was likely to test negative for all pathogens was also included. In round 2, participants used an online discussion board to debate the results of round 1. To the round 3 survey we added pathogens suggested by >1 participant and removed pathogens achieving <50% overall agreement in round 1, taking round 2 discussion into account. Due to anonymity, round 3 results could not be linked to round 1 results or round 2 comments.

Our primary outcome was the percent agreement of all participants recommending testing for a pathogen overall and in each domain (i.e., clinical management and public health) in round 3. Agreement for clinical management was differentiated based on immune status and compared using exact binomial tests. To reflect a scenario with no special considerations, overall agreement was calculated as the proportion of participants indicating a pathogen was important for either/both of clinical management for nonimmunocompromised children or public health purposes. If a participant skipped a question, they were excluded from the numerator *and* denominator when calculating agreement for that pathogen. Based on prior reviews of Delphi studies (30), we chose to classify high agreement as ≥80% and moderate agreement as ≥60% to <80%. To better contextualize results, we also calculated percent agreement by stakeholder role; specifically, the percent agreement of clinicians recommending testing for a pathogen for clinical management, and public health practitioners recommending testing for a pathogen for public health purposes.

## SUPPLEMENTAL MATERIAL

Supplemental material is available online only.
**SUPPLEMENTAL FILE 1**, PDF file, 0.8 MB.
**SUPPLEMENTAL FILE 2**, CSV file, 0.01 MB.

## ACKNOWLEDGMENTS

We acknowledge the contributions of advisory group members Samina Ali, Byron Berenger, Jia Hu, and Graham Tipples, who helped identify experts to invite and provided feedback on the initial survey design, and Ashton Chugh, who created the online survey tools. We also thank all the subject matter experts who provided their time and expertise to this study.

S.B.F. is supported by the Alberta Children's Hospital Foundation Professorship in Child Health and Wellness. G.A.M.T. was supported by a Canadian Institutes of Health Research (CIHR) Banting Postdoctoral Fellowship, Alberta Innovates Postgraduate Fellowship, and a University of Calgary Eyes High Postdoctoral Fellowship.

G.A.M.T., D.P., and S.B.F. have no conflicts of interest. P.I.T. serves as a member on the Data Safety Monitoring Board of Inmunova.

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
