## [Reviewer comments · Microbiology Spectrum]

Microbiology Spectrum

Enteric Pathogen Testing Importance for Children with Acute Gastroenteritis: A Modified Delphi Study

Gillian Tarr, Drew Persson, Phillip Tarr, and Stephen Freedman

Corresponding Author(s): Gillian Tarr, University of Minnesota

Review Timeline:

Submission Date:	May 20, 2022
Editorial Decision:	July 29, 2022
Revision Received:	August 19, 2022
Accepted:	August 31, 2022

Editor: Meghan Starolis

Reviewer(s): The reviewers have opted to remain anonymous.

Transaction Report:

DOI: <https://doi.org/10.1128/spectrum.01864-22>

July 29, 2022

Dr. Gillian A.M. Tarr
University of Minnesota
Environmental Health Sciences
MMC 807, Room 1240
420 Delaware St. SE
Minneapolis, MN 55455

Re: Spectrum01864-22 (Enteric Pathogen Testing Importance for Children with Acute Gastroenteritis: A Modified Delphi Study)

Dear Dr. Gillian A.M. Tarr:

Thank you for submitting your manuscript to Microbiology Spectrum. The review process has concluded and the reviewers have feedback which must be addressed before the manuscript can be accepted. When submitting the revised version of your paper, please provide (1) point-by-point responses to the issues raised by the reviewers as file type "Response to Reviewers," not in your cover letter, and (2) a PDF file that indicates the changes from the original submission (by highlighting or underlining the changes) as file type "Marked Up Manuscript - For Review Only". Please use this link to submit your revised manuscript - we strongly recommend that you submit your paper within the next 60 days or reach out to me. Detailed instructions on submitting your revised paper are below.

Link Not Available

Sincerely,

Meghan Starolis

Journals Department
Reviewer comments:

Reviewer #1 (Comments for the Author):

In the article by Tarr et al, the authors describe a modified Delphi study of various stakeholder groups (i.e., clinical experts, public health professionals and laboratory managers/directors) to determine the importance of different gastrointestinal pathogen testing in pediatrics. Three separate rounds of questions were used, after which there was a high level of agreement for testing of specific pathogens (i.e., Campylobacter, Escherichia coli O157 and other shiga-toxin producing E. coli, Salmonella, Shigella, Vibrio, Yersinia, norovirus and rotavirus). Additionally, the analysis also suggested that additional pathogen testing maybe needed in immunocompromised patients.

A. On lines 143-145, for the Round 3 analysis the authors list 26 stakeholders but Table 1, lists 15 individuals. Please clarify.

B. On line 260, it is mentioned that norovirus antigen is used within the United States. As far as I am aware, the only norovirus antigen test(s) that are FDA cleared are for outbreak investigation not routine clinical testing. Is that still correct? Also, I wonder if it should be mentioned that rotavirus antigen and molecular testing can result in positive results caused by the rotavirus vaccine. Caution should be taken with rotavirus testing in recently vaccinated individuals.

C. On line 306-307, the authors suggest that "selective suppression of results likely to indicate colonization", does create issues for the laboratory when positive results are known but not available to the clinical team. Ideally, test manufacturers would allow instrument suppression, so no-one is aware of the result, but that is not always available.

Reviewer #2 (Comments for the Author):

The authors performed a study to investigate the interest and priority in testing enteric pathogens from different subject matter experts. The contrary results in priority for testing certain pathogens between the clinical management and public health categories reflect the different needs and intents of testing. Overall, the data are interesting, although the "n" is relatively small. The presentation of the data could be improved as well.

Major comments

1. The main caveat of the study was that the data collected for public health were primarily given by clinicians and individuals not in the public health arena. Only 25% (5 of 20) of respondents in round 1 identified themselves as public health practitioners as their primary role, whereas only 1 of 15 in round 3 worked as a public health practitioner. I consider this survey mostly contributed by clinicians over lab professionals and public health practitioners (Table 1), thus questioning the validity of the public health opinions therein.

2. The calculation/definition of the overall agreement in Table 2 needs to be clarified and elaborated. The legend in line 457 was not clear. At first glance, the overall agreement seemed to be the highest percentage from the clinical management and public health groups, e.g. Salmonella, Shigella, Vibrio, but for other pathogens, the deduction of the overall agreement was not clear. In particular, the overall agreement in round 3 for rotavirus was 100%, while in both clinical management and public health groups the percentage agreements were all below 100%. The same applied to Sapovirus. Please explain.

3. Individual comments of respondents should be avoided in the result and discussion, as "cherry-picking" of comments could lead to bias. For example, lines 210-212, "...excluding it often enables us to move forward with other diagnoses..." The respondent also said "there is a sentiment that this is a pathogen" (omitted), contrasting the argument against the testing for Aeromonas "no clear evidence of pathogenicity" (appendix 3).

4. Ideally, clinicians want to rule out as many pathogens as possible for their diagnosis, but from the operation, budget, and reimbursement perspectives, the pan-pathogen testing could bring negative impacts. The authors should address more in the discussion.

5. The authors should address the fact that diagnostic tests are more available for certain pathogens (enteric bacteria overall, and parasites) over the others (enteric viruses), thus possibly skewing the feedback of the respondents. One example is that the PCR specific for Aeromonas is very rare in the clinical field. "When you don't look for it, you don't find it". The incidence and impact of Aeromonas infection may be underestimated.

Minor comments

1. Statistics should be applied in the comparison between round 3 IC vs NIC.

2. In Table S1, both Cryptosporidium and Vibrio had 100% agreement in round 1 and round 3. Much focus was on the latter but not the former throughout the manuscript. Why was that?

Staff Comments:

Preparing Revision Guidelines

To submit your modified manuscript, log onto the eJP submission site at <https://spectrum.msubmit.net/cgi-bin/main.plex>. Go to Author Tasks and click the appropriate manuscript title to begin the revision process. The information that you entered when you

first submitted the paper will be displayed. Please update the information as necessary. Here are a few examples of required updates that authors must address:

Please return the manuscript within 60 days; if you cannot complete the modification within this time period, please contact me. If you do not wish to modify the manuscript and prefer to submit it to another journal, please notify me of your decision immediately so that the manuscript may be formally withdrawn from consideration by Microbiology Spectrum.

Reviewer comments:

We appreciate the helpful comments from the reviewers and have responded to each below. Line numbers refer to the clean, unmarked revised version of the manuscript.

Reviewer #1 (Comments for the Author):

In the article by Tarr et al, the authors describe a modified Delphi study of various stakeholder groups (i.e., clinical experts, public health professionals and laboratory managers/directors) to determine the importance of different gastrointestinal pathogen testing in pediatrics. Three separate rounds of questions were used, after which there was a high level of agreement for testing of specific pathogens (i.e., Campylobacter, Escherichia coli O157 and other shiga-toxin producing E. coli, Salmonella, Shigella, Vibrio, Yersinia, norovirus and rotavirus). Additionally, the analysis also suggested that additional pathogen testing maybe needed in immunocompromised patients.

A. On lines 143-145, for the Round 3 analysis the authors list 26 stakeholders but Table 1, lists 15 individuals. Please clarify.

Participants were able to indicate up to two stakeholder roles; e.g., someone could be both a clinician and a content expert. We have modified the sentence in Results to make this clearer (lines 144-146).

B. On line 260, it is mentioned that norovirus antigen is used within the United States. As far as I am aware, the only norovirus antigen test(s) that are FDA cleared are for outbreak investigation not routine clinical testing. Is that still correct? Also, I wonder if it should be mentioned that rotavirus antigen and molecular testing can result in positive results caused by the rotavirus vaccine. Caution should be taken with rotavirus testing in recently vaccinated individuals.

We believe the reviewer is correct on both points. We have added comments on both to the Discussion (lines 255-257 and 263).

C. On line 306-307, the authors suggest that "selective suppression of results likely to indicate colonization", does create issues for the laboratory when positive results are known but not available to the clinical team. Ideally, test manufacturers would allow instrument suppression, so no-one is aware of the result, but that is not always available.

The reviewer makes an important point. We have added this caution (lines 323-326).

Reviewer #2 (Comments for the Author):

The authors performed a study to investigate the interest and priority in testing enteric pathogens from different subject matter experts. The contrary results in priority for testing certain pathogens between the clinical management and public health categories reflect the different needs and intents of testing. Overall, the data are interesting, although the "n" is relatively small. The presentation of the data could be improved as well.

Major comments

1. The main caveat of the study was that the data collected for public health were primarily given by clinicians and individuals not in the public health arena. Only 25% (5 of 20) of respondents in round 1 identified themselves as public health practitioners as their primary role, whereas only 1 of 15 in round 3 worked as a public health practitioner. I consider this survey mostly contributed by clinicians over lab professionals and public health practitioners (Table 1), thus questioning the validity of the public health opinions therein.

Our goal was to elicit the opinions of pediatric gastroenteritis subject matter experts from multiple backgrounds. We agree with the reviewer that the number of respondents specifically working as public health practitioners, particularly in Round 3, was suboptimal. However, we believe there is still value in

the opinions of pediatric gastroenteritis content experts from other domains. Many of the non-public health practitioner participants conduct research that informs public health policy and practice, oversee public health laboratories, and collaborate closely with public health departments on a wide range of initiatives. While we do not know individual responses by qualifications due to the need for anonymity, less than half of respondents to each survey indicated their primary role as clinician, and >70% of participants in rounds 1 and 3 indicated two stakeholder roles. Based on this, we believe participant representation is closer to the broad subject matter expertise we were seeking than a simple comparison of the number of clinician and public health practitioner respondents would suggest. This is bolstered by the large degree of concordance between the results incorporating all respondents and those including only public health practitioners. We have added a more detailed discussion of this issue on lines 281-301. This includes an expanded discussion of the implications of this limitation, whereas it had previously been mentioned but not explained in depth in the limitations paragraph.

2. The calculation/definition of the overall agreement in Table 2 needs to be clarified and elaborated. The legend in line 457 was not clear. At first glance, the overall agreement seemed to be the highest percentage from the clinical management and public health groups, e.g. Salmonella, Shigella, Vibrio, but for other pathogens, the deduction of the overall agreement was not clear. In particular, the overall agreement in round 3 for rotavirus was 100%, while in both clinical management and public health groups the percentage agreements were all below 100%. The same applied to Sapovirus. Please explain.

The figure footnote has been clarified. It now reads, "Overall agreement was calculated as the proportion of participants indicating a pathogen was important for either/both non-IC clinical management and/or public health purposes (i.e. the proportion of participants indicating importance for at least one of these purposes)." (lines 514-515)

3. Individual comments of respondents should be avoided in the result and discussion, as "cherry-picking" of comments could lead to bias. For example, lines 210-212, "...excluding it often enables us to move forward with other diagnoses..." The respondent also said "there is a sentiment that this is a pathogen" (omitted), contrasting the argument against the testing for Aeromonas "no clear evidence of pathogenicity" (appendix 3).

We used the quotes as evidence of the themes identified in the qualitative data. We have attempted to follow best practices of quotations in qualitative research in presenting these data (e.g. from Lingard, *Perspect Med Educ.* 2019 Dec; 8(6): 360–364). Specifically, we have included quotes only where they are relevant to the point being made in the paper; the example the reviewer provides is within the Results section, *Disagreement Existed Regarding Utility of Testing to Inform Differential Diagnosis*, and was therefore seen as relevant. We have also sought to keep quotes succinct, which as the reviewer indicated, in this case meant removing information that was not directly relevant to the theme it was included in reference to. We do not believe this amounts to cherry-picking the data in a biased way, particularly as the section pertains to lack of agreement and highlights the low number of individuals ranking *Aeromonas* testing as a priority. The two quotes in the subsequent paragraph further provide rationale behind the two sides of the differential diagnosis disagreement.

We have reviewed all quotes used in the paper to ensure all follow these best practices and subsequently removed one from the Discussion (lines 253-255). If the journal prefers that quotations from qualitative data not be included, we will remove the rest at the editor's request.

4. Ideally, clinicians want to rule out as many pathogens as possible for their diagnosis, but from the operation, budget, and reimbursement perspectives, the pan-pathogen testing could bring negative impacts. The authors should address more in the discussion.

The reviewer makes an excellent point. We have added a discussion of this (lines 313-319).

5. The authors should address the fact that diagnostic tests are more available for certain pathogens (enteric bacteria overall, and parasites) over the others (enteric viruses), thus possibly skewing the feedback of the respondents. One example is that the PCR specific for *Aeromonas* is very rare in the clinical field. "When you don't look for it, you don't find it". The incidence and impact of *Aeromonas* infection may be underestimated.

The reviewer makes an important point. We have added a discussion of this limitation (lines 336-342).

Minor comments

1. Statistics should be applied in the comparison between round 3 IC vs NIC.

We have added this analysis as requested and modified language in multiple places to match the results (Table S3, lines 202-203, 236, and 345).

2. In Table S1, both *Cryptosporidium* and *Vibrio* had 100% agreement in round 1 and round 3. Much focus was on the latter but not the former throughout the manuscript. Why was that?

This is because there was already high overall agreement for *Vibrio* but only moderate overall agreement for *Cryptosporidium* (Table 2). The emphasis placed on *Cryptosporidium* in the Discussion was 1) as an example of a pathogen that may possibly have been 'elevated', so to speak, from moderate to high agreement had there been more public health-oriented SMEs among the participants (see the discussion now on lines 297-299), and 2) as one of the pathogens for which the domain-specific stakeholders differed from the overall participants (the paragraph on lines 303-309).

August 31, 2022

Dr. Gillian A.M. Tarr
University of Minnesota
Environmental Health Sciences
MMC 807, Room 1240
420 Delaware St. SE
Minneapolis, MN 55455

Re: Spectrum01864-22R1 (Enteric Pathogen Testing Importance for Children with Acute Gastroenteritis: A Modified Delphi Study)

Dear Dr. Gillian A.M. Tarr:

After review of the responses to the reviewers, I have found your revisions had adequately addressed all questions and concerns. Your manuscript has been accepted, and I am forwarding it to the ASM Journals Department for publication. You will be notified when your proofs are ready to be viewed.

Sincerely,

Meghan Starolis
Editor, Microbiology Spectrum

Journals Department
Supplemental Material: Accept
Supplemental Dataset: Accept